# Do Clinical Trials Meet Current Care Needs? Views of Digestive Oncology Specialists in Galicia (Spain) Using the Delphi Method

**DOI:** 10.3390/healthcare9060665

**Published:** 2021-06-03

**Authors:** Ana Fernández Montes, Nieves Martinez-Lago, Juan de la Cámara Gomez, Elena María Brozos Vázquez, Sonia Candamio Folgar, Marta Carmona Campos, Antía Cousillas Castiñeiras, Marta Covela Rúa, Elena Gallardo Martín, Paula González Villarroel, Begoña Graña Suarez, Mónica Jorge Fernández, María Luz Pellón Augusto, Guillermo Quintero Aldana, Carlos Romero Reinoso, Mercedes Salgado Fernández, Francisca Vázquez Rivera, Ana Ayuso Álvarez, Dante R. Culqui, José Carlos Méndez Méndez

**Affiliations:** 1Complexo Hospitalario Universitario de Ourense, 32005 Ourense, Spain; afm1003@hotmail.com (A.F.M.); mercedes.salgado.fernandez@sergas.es (M.S.F.); secretariagitud@gmail.com (R.-s.W.G.); 2Medical Oncology Department, University Hospital A Coruña, 15006 A Coruña, Spain; nieves.purificacion.martinez.lago@sergas.es (N.M.-L.); begona.grana.suarez@sergas.es (B.G.S.); 3Complexo Hospitalario Universitario de Ferrol, 15405 Ferrol, Spain; jcamarakq@gmail.com (J.d.l.C.G.); maria.luz.pellon.augusto@sergas.es (M.L.P.A.); 4Complexo Hospitalario Universitario de Santiago de Compostela, 15706 Santiago de Compostela, Spain; elena.maria.brozos.vazquez@sergas.es (E.M.B.V.); sonia.candamio.folgar@sergas.es (S.C.F.); francisca.vazquez.rivera@sergas.es (F.V.R.); 5Hospital Universitario Lucus Augusti, 27003 Lugo, Spain; marta.carmona.campos@sergas.es (M.C.C.); marta.covela.rua@sergas.es (M.C.R.); guillermo.quintero.aldana@sergas.es (G.Q.A.); 6University Hospital Pontevedra, 36002 Pontevedra, Spain; antia.cousillas.castiñeiras@sergas.es; 7University Hospital Vigo, 36213 Vigo, Spain; elena.gallardo.martin@sergas.es; 8Hospital Álvaro Cunqueiro, 36213 Vigo, Spain; paula.gonzalez.villarroel@sergas.es (P.G.V.); monica.jorge.fernandez@sergas.es (M.J.F.); 9Hospital Povisa, 36211 Vigo, Spain; cromero@povisa.es; 10Instituto de Salud Carlos III, 28029 Madrid, Spain; ayusoalvarez@gmail.com; 11Sociosanitario (Elderly Hospital) Isabel Roig, 08030 Barcelona, Spain; 12Health National School Carlos III Health Institute, 28029 Madrid, Spain; 13Sanatorio Nosa Señora dos Ollos Grandes, 27001 Lugo, Spain; jcmendez.m@gmail.com

**Keywords:** views digestive oncology specialists, Delphi method

## Abstract

**Background:** In recent years, abundant scientific evidence has been generated based on clinical trials (CT) in the field of oncology. The general objective of this paper is to find out the extent to which decision making is based on knowledge of the most recent CT. Its specific objectives are to pinpoint difficulties with decision making based on the CT performed and find out the motivations patients and clinicians have when taking part in a CT. **Methodology:** Combined, prospective study, based on the Delphi method. A lack of correspondence between the people who take part in CT and patients who come for consultation has been identified. A need for training in analysing and interpreting CT has also been identified and a lack of trust in the results of CT financed by the pharmaceutical industry itself has been perceived. **Conclusions:** There is a difficulty in selecting oncological treatment due to the lack of correspondence between the patients included in the CT and patients seen in consultation. In this process, real world data studies may be highly useful, as they may provide this group with greater training in interpreting CT and their results.

## 1. Introduction

Over the last decade, many publications have questioned the usefulness of CT. John P. A. Ioannidis raises the need to take an appropriate clinical approach to them and consider their usefulness when the results bring about a change in therapeutic decision making [1].

By 2014, approximately one million CT and tens of thousands of systematic reviews had been published, the majority of which were not clinically useful. It is estimated that 85% of the billions invested in research each year goes into CT [2].

The majority of CT are designed with the aim of demonstrating the efficacy and safety of medications in order for the pharmaceutical industry to obtain medium/long-term regulatory approvals (clinical developments from early phases to phase III or final, pivotal or registration phases).

Many authors deem that if a CT is very flexible in its design, definitions or analysis, its results may be less rigorous [3].

In order to have better control during the design and performance of a CT, tools have been created, such as that produced by Cochrane, which seeks to set out minimum criteria to determine the quality of the articles analysed, and both the internal and external validity of CT [4] is assessed in order to do this. In oncology there are constant therapeutic advances and, in a matter of just a few years, there has been a shift from cytotoxic drugs to monoclonal antibodies as specific therapeutic targets or immune response modulating drugs to treat a tumour. In addition, in some cases there are different combinations of all of these, which makes them more complicated to analyse.

In this speciality, there are major advances in clinical treatment for multiple types of tumour with different histologies, specific biomarkers, drug combinations and mechanisms of action. Generally speaking, precision medicine has allowed a great advance in the efficacy and safety of the various therapeutic strategies. There is thus a great amount of information concerning new drugs tested in populations that do not adequately represent the type of patients actually seen by specialists in their day-to-day clinical practice. On many occasions, this means that the results of CT are extended to special populations that are not included in them. For this reason, some authors suggest using Real World Data Studies (RWD); the term real world data (RWD) refers to population-level data obtained from cancer registries and not patient information extracted from a study [5]. The Food and Drug Administration (FDA) has defined RWD as data relating to patient health status and/or care records that are real and that are collected during the patient’s care under real, rather than ideal conditions. This information comes from the electronic clinical history and from administrative records that the patient was added to during clinical care. [6].

This problem raises questions concerning the methodology of CT. Just how prepared are we to deal with the new methodological designs necessary to analyse new treatments in all populations? Do CT represent the population we want to treat? Researchers themselves are aware of these aspects, but how do they perceive and tackle problems with CT in day-to-day clinical practice?

The RIGhT-sens study (Delivering a Right and Individualized Digestive Tumor Treatment) is intended to show the perception of medical oncology specialists who belong to the Galician Digestive Tumour Research Group (*Grupo Gallego de Investigación en Tumores Digestivos de Galicia*—GITuD). The aim of this group is to raise medical awareness of the influence of CT on the handling of patients with digestive tumours in Galicia and in Spain.

The general objective of this study is to find out the extent to which oncology specialists are aware of CT, both their quality and their interpretation, as well as how they use them when deciding on one treatment or another.

## 2. Methodology

### 2.1. Type of Study

Combined (qualitative and quantitative), prospective study using a modified Delphi method.

A several-round Delphi study was proposed. However, due to information saturation the study was completed in the first round.

The Delphi method is a systematic forecasting method, the objective of which is to obtain a consensus based on discussion by experts (oncologists) through an iterative process. It is used when there is scarce empirical evidence, the data are vague, and subjective factors predominate [7].

This methodology has high reliability, flexibility, dynamism and validity because it allows for anonymous participation, a heterogeneous group of experts, iteration and prolonged feedback between participants, and avoids problems of representativeness and control of the discourse by some people over others [7]. In addition, there is evidence concerning the certainty it generates in decision making, since this responsibility is shared by all of the participants [8].

Stage 1:

Identifying participants:

A working group (RIGhT-sens group) proposed a list of probable experts who could take part in the study, who were sent an invitation to take part in the study. They had all worked as part of a clinical trial. Finally, 14 experts from different hospitals in Galicia accepted.

All of the experts selected were oncological physicians who had participated in a clinical trial at least once. The majority of those surveyed had at least two years’ experience in treating gastric cancer patients.

Sampling: Intentional non-probabilistic sampling was used.

Preparation of the questionnaire:

The panel of experts was commissioned with identifying the questions to be assessed (32 questions) in different areas to be explored. Based on these, the following areas were identified:

A questionnaire was created with semi-structured and open questions that tackled different topics with the following structure:

I. In the Case of There Being No Clinical Trials in Progress in Your Hospital. Exploration of medical problems in the use of treatments after the results of the clinical trial have been obtained.

II. In the Case of There Being Clinical Trials in Progress in Your Hospital, How Do They Affect Your Usual Practice?

III. Knowledge of Clinical Trials

The issues tackled were: problems identified in performing clinical trials (in the workplace and in patient recruitment); assessment of the usefulness and interest of clinical trials; assessment of the risks versus benefits of participating; knowledge of the organisations funding the trial; and, if any, the type of compensation patients and clinics received for participating. They were also asked about their hospital’s logistical capabilities to handle the performance of clinical trials (specialised personnel, equipment and time to perform it); the patient recruitment process; and, lastly, they were asked about their degree of participation and knowledge of clinical trials.

### 2.2. Collecting Information

An invitation was sent to the experts to be interviewed. If the experts agreed to participate, they were sent a link to a virtual platform containing information about the trial and the questionnaires.

### 2.3. Participant Selection

The working group (RIGhT-sens) sent an invitation to various oncology specialists who work in different hospitals in Galicia, Spain, to participate in the study. The specialists who agreed to participate received an online questionnaire. The specialists were selected due to working in at least one of the eight main hospitals in Galicia, Spain.

The interview questionnaire was drawn up by four experts from GITuD and filled in by another 14 oncology experts in the Galician Health Service (SERGAS) who work in eight hospitals in Galicia: Complexo Hospitalario Universitario de Orense (CHOU) (one), Complexo Hospitalario de Pontevedra (CHOP) (two), Hospital Clínico Universitario de Santiago (CHUS) (three), Complejo Hospitalario Universitario A Coruña (CHUAC) (one), Complexo Hospitalario Universitario de Ferrol (CHUF) (one), Hospital Universitario Lucus Augusti (HULA) (three), Complejo Hospitalario Universitario de Vigo (CHUVI) (two), and Hospital Povisa (one).

### 2.4. Study Time

The survey was conducted online between 11 November and 18 December 2019.

Stage 2: Type of analysis and consensus:

The questionnaire, made up of closed and open questions, made it possible to conduct a descriptive analysis (quantitative method) and a discourse analysis (qualitative method).

Selection of questions for the test:

Questionnaire preparation: this was carried out with the RiGhT-sens working group. The RIGhT-sens group was shown each of the probable questions together with explanatory text. They were then asked to vote to maintain, remove or modify the question, or state that they had no opinion. We used categorical response options to ensure that the people taking the survey fully understood the consequences of their votes, to clarify the interpretation and to ensure that the results could be processed in such a way as to establish a final list of questions at the end of the study.

First Round: In the first round, two types of qualitative and quantitative analysis were carried out, as described below:

Quantitative analysis: in order to analyse the first round, SPSS statistical software was used to perform a descriptive analysis of the closed-ended answers (questions in which an alternative was selected).

Qualitative analysis: Analysis of the contents of the qualitative questions was carried out.

### 2.5. First Round Test

Once the questions had been selected, a link to the tests was sent. All of the questions had options to reduce the response margin and a free text option to add supplementary information if it was necessary to add it.

Consensus: The frequency distribution of each variable was then calculated and the highest percentage for each answer was identified.

In a second stage, the mode of the highest percentage of all answers was calculated in order to establish a consensus value, which was finally set at 64.29% for the majority of variables analysed. We chose the mode in order to represent the highest number of responses and prevent our observations being affected by extreme values. When we verified the value established, we also confirmed that the consensus value was within the ranges proposed in other research, which mention that a consensus level can vary from 51% to 100% and that a level of 100% is unlikely to be reached [9,10].

There was considered to be consensus when there was more than 64% agreement with the identified responses. For questions with a lower level of consensus, the analysis was complemented with qualitative analysis (content analysis) in accordance with any of the available actions. The RIGhT-sens teams analysed the responses to the first round questionnaire. “No opinion” responses were excluded from the agreement percentage calculations. The initial plan was that whenever there was no consensus on maintaining or removing a question, it would not be included in the questionnaire in the next round. However, such discrimination was not necessary since there was not sufficient information for a second round. In cases in which the panel did not reach at least 64% agreement to maintain or remove a question, we examined the comments and extrapolated the most important idea.

After the initial analysis, it was identified that a high percentage of questions reached a consensus level higher than 64%. The responses with a lower consensus percentage were supplemented with qualitative analysis provided in open questions.

The contents were analysed in parallel by two researchers (an epidemiologist and a health anthropologist) in order to avoid intersubjectivity and triangulate the results. Once it had been collected, the information was analysed until the discourse was saturated.

Consensus concerning the results and their internal consistency made it unnecessary to perform a second round of consultation.

Communication of information to participants:

All of the information was shared with the participants once the analysis had been completed.

## 3. Results

The results were obtained from analysing 30 of the 32 questions in the questionnaire, since two of them did not offer significant information (question 5, which referred to the most important weaknesses in extrapolating the results of clinical trials to the work environment, and question 26, which asked about the specialists’ prior participation in clinical trials).

Following the quantitative analysis, with the results shown in Table 1, and the qualitative analysis, which gathers the testimony of the experts reviewed in their cognitive, social and hospital context, the main finding observed in the study was a lack of correspondence between the people who take part in CT and the patients who come for consultation. So, almost half of the oncologists (43%) did not consider the information from them to be sufficient for therapeutic decision making concerning their patients.

In cases in which a CT was not being conducted in their hospital (see the testimony in Table 2), the information from CT was not considered sufficient for decision making, since the indications in the CT are individualised according to the patient to be treated (testimony 1 [T1]), and it was also stated that it is difficult to find specific evidence for some diseases or certain treatment stages (T2). Therefore, it was observed that CT are not adapted to the needs of their patients (T3), their state of health, or the contextual realities in which they work (T4, T5). So, they all agreed with the need for real world data (RWD) studies as a complement to CT to assess the efficacy of treatment (T6).

Generally speaking, the majority of the experts stated that it was necessary to confirm probable diagnoses in the real population or a population with similar characteristics to those they saw in consultation.

In cases in which there was a CT in the hospital (see the testimony in Table 3), more than half of the interviewees (65%) denied receiving any kind of compensation for including patients in a CT, but if there was any financial compensation, they mentioned that the researchers do not receive it directly (T7).

With regard to the usefulness of CT, it was stated that they were useful in clinical practice and using them as a basis offered more benefits than risks. In addition, they made it possible to find new solutions to existing problems and the act of studying those problems in their own patients was an extra incentive (T8).

It was also detected that CT could change the paradigms of clinical practice in the short to medium term, such as when a new treatment alternative is offered (T9).

It was argued that the interest in performing a CT was normally due to patient need (T10), although on some occasions it was due to private interests.

In spite of the majority considering that personnel were qualified to be monitors or responsible for supervising a CT, some of them had doubts about this (T11, T12) and said that, sometimes, monitors were not sufficiently trained, due to several factors: the profile of the people performing the task (inexperienced young people) and poor working conditions, which brought about a continual change in personnel (T13, T14).

With regard to logistics for the performance of CT, the majority mentioned that they have a data manager and a management unit in their centre (71%), but they complained about a lack of support from the hospital, in spite of the fact that it could receive benefits from performing this kind of activity (T15). They also recognised that external groups were entrusted with performing the analysis (T16, T17).

In the case of a CT being performed in a hospital, no change or improvements in the logistics or infrastructure of the hospital were perceived as a consequence of it, so it also had no effect on recruitment or inclusion of patients (T18).

Among the factors that may influence the participation of patients in CT, they mentioned the patients’ socio-economic situation (T19) and the number of tests to be performed on patients and their comorbidities (T20, T21).

With regard to the criteria for selecting patients, a lack of alternatives or treatment is what motivated most of the specialists to include their patients in a CT, due to considering it to be the best option for them (T23, T24). Almost all of them had recommended a CT at some time, since they were convinced that it was the most beneficial therapeutic option (86%), based on the medical principle of *primum non nocere* (T25, T26). They stated that they did not consider themselves conditioned by financial incentives (T22).

More than half of the interviewees (64%) commented that it is difficult to find the ideal patient to include in a CT and that they found it hard to recruit patients who met all of the criteria and, also, that they were not likely to accept.

One of the reasons they all recognised as a motivation for performing a CT was the importance of being the author of a CT, since it benefits them from the viewpoint of their CV (T27), and because being an author of a publication is a form of recognition of their work and a reward for it (T28, T29). Although, the majority did not consider it to be an indispensable requirement (T30), instead the important thing was to offer a possible better treatment alternative to patients (T31).

More than half of them had issued a publication prior to their first CT (65%); almost 30% of these had published during the resident stage.

One influential factor in the decision to participate in a clinical trial, according to 43% of the interviewees, was the financial incentive (T32, T33, T34), but this had a different influence depending on whether they were patients or professionals. In the case of patients, their poor economic situation was mentioned and, among health professionals, the main reason was to provide a better treatment option to the patient. Almost all of them agreed that CT are useful when they offer new alternatives to patients (93%) and when they provide data about treatment sequences or subpopulations (T35, T36).

Regarding their knowledge of CT (see the testimonies in Table 4), the majority were aware of biological markers and some research techniques such as biases, although only 57% of the specialists managed to mention selection bias and very few mentioned a different type of bias (15%). Some of them were even mistaken regarding evaluations of biases.

When they were asked about which parts of a CT are analysed before deciding to take part in a study, the specialists mentioned the following in order of priority:Characteristics of the population studied (93%).Ethical considerations of the study (86%).Representativeness of the study (64%).

Almost half of them admitted that they did not have sufficient abilities to analyse a CT (43%), and 50% of these recognised that they were lacking in statistical skills.

It was found that just 29% of the specialists analyse clinical protocols before prescribing new drugs in order to find out whether the population is similar to the one being treated, or if the study was well-produced methodologically; 64% do this sometimes.

Finally, some of them said that during the performance of a CT they had identified groups applying pressure for the use of a particular treatment (T37, T38).

## 4. Discussion

More than 40% of the specialists who did not have a CT in progress in their hospital mentioned that the information about the CT they take as a basis is insufficient. It must be applied with modifications when it is to be used with the patients they are each treating, since the results of a CT cannot be applied to people who are outside the scope of the sample used [3]. There is clear concern among specialists who treat the elderly population in Spain, where life expectancy increased to 86 for women and 80 for men in 2016 [11]. All of the specialists were of the opinion that real world data studies must be prioritised as a measure to be able to apply the proposed treatments to real patients.

The majority of interviewees who had a CT in their hospital did not receive financial benefits, in spite of these trials requiring special logistics and different levels of financing; it is important for the benefit always to be higher than the cost of carrying it out [12].

The main interest of a CT is to respond to a patient’s need, although sometimes CT are due to researchers’ interests. This is important because CT must be useful in themselves, without the underlying interests mattering; otherwise we would be moving away from patient-centred research [13].

In order to achieve this, it is important for monitors to be adequately trained, since a lack of training in professionals involved in clinical trials affects the development of the EC itself [1], and it has been observed in this context that many doctors are underemployed as volunteers [14].

In addition, the majority stated that sometimes it is the pharmaceutical companies that analyse the information from CT, and this creates suspicion among professionals. This fear could disappear if independent funding were found and there were no conflicts of interest with the pharmaceutical industry.

When performing a CT, one of the factors that may influence a patient’s decision to participate is his/her socio-economic situation. This is in keeping with the results of other studies on adherence to treatment, in which an association was identified between financial difficulties and a decrease in attendance by patients to receive treatment [15].

Moreover, one of the criteria for the specialist not including a patient in a CT is their lack of understanding, because the patient will not follow the experimental treatment or will do so irregularly. Another factor is the patient’s complexity, the treatment’s side-effects and the patient’s lack of motivation [3].

The specialists commented that they did not receive direct benefits for including a patient in research, at least no financial benefits, and only rarely academic benefits. The majority commented that they had not felt forced to include patients in a CT and had recommended a CT, convinced that it was the most appropriate therapeutic option. However, between 2014 and 2016, the Food and Drug Administration (FDA) in the United States approved 47 antineoplastics, only nine (19%) of which complied with the standards of the American Society of Clinical Oncology to consider the drug as having a significant clinical benefit [16].

There were 64.29%, of the opinion that it was very difficult to find an ideal patient to participate in a CT. In spite of this, many epidemiologists think that it is only through randomised controlled CT [17,18,19] or meta-analyses [20,21] that reliable results can be proven, because it is believed that the more flexible the designs or analysis methods are, the less trustworthy the results are [7].

Regarding the knowledge that the specialists have of CT, 51% of the interviewees mentioned that they were aware of selection bias. Although the risk of this bias appearing is intimately linked to clinical research [22], few researchers are aware of the biases from which they may suffer. The most well-known selection biases are Berkson’s or Neymann’s bias, among others [23,24], but there are other groups such as information, measurement, etc., which were not mentioned. Regarding this, one study showed that out of 704 articles, only 514 (73%) had the help of an expert in methodology and statistics. The rejection rate was 71% for those that did not have these experts and 57% for those that did [25].

It is concerning that 46% considered that they do not have sufficient abilities to analyse a CT and that 50% recognised that they were lacking in statistical skills. There are those who are of the opinion that an oncologist should be knowledgeable in his/her field and not an expert in statistics. Others consider that any researcher knows enough about statistics [3]; this would explain some errors in the use of statistical evidence in CT [26]. Both forms of knowledge, clinical and statistical, must go hand-in-hand in a CT.

It is therefore important for hospitals to have a research department, which should have a statistical team or committee of experts who are responsible for helping and supporting people interested in conducting research with cancer patients.

Some of the interviewees mentioned a kind of pressure from oncologists and pharmaceutical companies; regarding this, sponsored research initiatives should be patient-focused [27] and professionals should accept the challenges of producing CT. Patient empowerment groups could help avoid the influence of biases in the future [1]. Patient associations can also exert pressure. One example is the case of interferon beta-1b, which was initially used to treat Multiple Sclerosis. It was approved without complying with the necessary specifications [28] for safety and efficacy due to patient pressure. Another example is azidothymidine, which was also authorised due to pressure from patient associations and excess mortality [29]. The FDA implemented a fast-track process to accelerate the approval of some protocols [30].

## 5. Limitations

Despite the small size of the population studied, and despite the fact that it would seem intuitive that a larger study might be required to identify a sample from areas of Spain other than Galicia, we are aware that, from a qualitative point of view, causal inference is not an important limitation. We therefore consider that these findings are valid and could be very useful for other oncologists both in Spain and around the world. Moreover, qualitative research allows us to identify findings that it would be difficult to demonstrate from a quantitative point of view.

## 6. Conclusions

Decision making concerning oncological treatments of therapies based on CT is hindered by the lack of correspondence between the patients included in the CT and patients seen in consultation. To make up for this deficiency, inclusion of real world data studies is proposed.

Aiding the professional progression of their research career is an incentive for clinicians to participate in a CT.

The need for training in critical reading and analysis of CT for this group has also been revealed.

One way of ending the distrust generated by CT performed by the pharmaceutical industry is for independent financial bodies to develop these kinds of studies.

This study shows that, often, the decisions in some situations are decisions that should be individualised for each patient because no protocols are identified that are adjusted to such specific situations at the time of treating the oncology patient.

## Figures and Tables

**Table 1 healthcare-09-00665-t001:** Quantitative analysis of the perception of the usefulness of clinical trials Galicia, 2020.

	Yes		No		Total
	*n*	%	*n*	%	
**In the case of there being no clinical trials in progress in your hospital**					
**Exploration of medical problems in the use of treatments after the results of the clinical trial have been obtained**					
1. Did you find sufficient information on the results of existing clinical trials to take adequate decisions for the treatment of the majority of your patients? Explain your answer.	8	57.14	6	42.86	14
2. Do you consider the results of existing clinical trials to be suited to the needs of your patients? Explain your answer.	5	35.71	9	64.29	14
3. Do you consider the study populations in clinical trials that you have reviewed and used in the treatment of your patients to be similar to the population you treat? Explain your answer.	5	35.71	9	64.29	14
4. Do you consider that more real world data (RWD) studies are necessary in order to assess the efficacy of a treatment as a complement to the pivotal clinical trial? Explain your reasons.	14	100.00	0	0.00	14
**In the case of there being clinical trials in progress in your hospital, how do they affect your usual practice?**					
General characteristics					
6. Do you receive financial or non-financial (academic or other) compensation for including patients in clinical trials? Specify the type of compensation.	5	35.71	9	64.29	14
7. Have you found that often the risks of clinical trials in which you have taken part are greater than the benefits the patient may receive? Explain what you have observed.	1	7.14	13	92.86	14
**Logistics for handling clinical trials**					
10. Do you consider that the monitors of clinical trials are sufficiently trained to control the characteristics of the clinical trials or have you observed that there was not adequate control? Explain your answer.	11	78.57	3	21.43	14
11. Does your centre have a clinical trial management unit or a data manager?	10	71.43	4	28.57	14
13. Have you observed that, when an important clinical trial is being performed in your hospital, there are improved logistics for monitoring equipment or treatment equipment that will be subsequently used for patients? What equipment?	3	21.43	11	78.57	14
14. If there is, or already was, new equipment donated by pharmaceutical companies, new units implemented by pharmaceutical companies or any additional benefit for the hospital, do you consider that receiving these benefits conditions the recruitment or inclusion of patients in the study? Explain your answer.	5	35.71	9	64.29	14
**Participant selection**					
17. Do you think that the financial or academic benefit received for patients included in the study affected any of your criteria for including a patient? You may give an example.	2	14.29	12	85.71	14
18. Do you consider that sometimes you have been forced to include patients in a clinical trial because there are no other treatment alternatives in your hospital? You may give an example.	4	28.57	10	71.43	14
19. When you recommend a patient to take part in a study, are you convinced that it is the most beneficial therapeutic option for the patient? Why?	12	85.71	2	14.29	14
**Doctors’ motivation**					
21. Do you think it is important to participate as an author of the clinical trials on which you have collaborated? Why?	14	100.00	0	0.00	14
22. Is it a requirement for you to be allowed to be an author in order to participate in a clinical trial? Why?	2	14.29	12	85.71	14
23. Before taking part in the first clinical trial, had you issued publications? Why?	9	64.29	5	35.71	14
24. As a medical professional, do you think that the financial incentive influences the decision to take part in a clinical trial? Why?	6	42.86	8	57.14	14
25. Do you consider the clinical trials in process will be useful for implementation in the future? Why?	13	92.86	1	7.14	14
**Knowledge of clinical trials**					
27. Do you know how biological markers are used in clinical trials? Mention a biological marker used in clinical research.	12	85.71	2	14.29	14
28. Do you know any kinds of biases that may affect a clinical trial?	12	85.71	2	14.29	14
30. Do you consider yourself capable of analysing a clinical trial and deciding if it is well-produced or has biases? Why?	8	57.14	6	42.86	14
32. Have you identified any groups that have exerted pressure for the use of a particular treatment?	5	35.71	9	64.29	14
The questions concerning only qualitative analysis are not included in this table.					

**Table 2 healthcare-09-00665-t002:** Qualitative analysis of the assessment of usual practice in the case of there being no clinical trials in progress in your hospital. Galicia, 2020.

Questions	Qualitative Textual Quote
1. Did you find sufficient information on the results of existing clinical trials to take adequate decisions for the treatment of the majority of your patients? Explain your answer.	*“The patients included in the trials match the clinical characteristics of a percentage (30–40%) of the patients I see day to day, so (although the decisions are based on the trials) the indications and recommendations are adapted and personalised to each patient for the other 70–60%.”* *(T1).* *“The low evidence in many situations is a problem (for example, bile duct, second or third lines in multiple neoplasms, etc.); moreover, the absence of comparison between therapeutic options (adjuvant therapy in gastric cancer, first line in pancreatic cancer, etc.) is also a problem.”* *(T2).*
2. Do you consider the results of existing clinical trials to be suited to the needs of your patients? Explain your answer.	*“The majority of the patients included in clinical trials are not comparable to those we see in our day-to-day practice.” (T3).*
3. Do you consider the study populations in clinical trials that you have reviewed and used in the treatment of your patients to be similar to the population you treat? Explain your answer.	*“I work in an area with an elderly population, which is not usually included in clinical trials.” (T4).* *“The patients in clinical trials are selected for their general state of health and comorbidities and a large number of patients from the real population are left out.” (T5).*
4. Do you consider that more real world data (RWD) studies are necessary in order to assess the efficacy of a treatment as a complement to the pivotal clinical trial? Explain your reasons.	*“RWD studies allow the results to be verified in the real population and not in an ideal situation.” (T6).*

T: testimony. The testimonials (T) represent the opinions of the experts; more than one opinion may have been issued by an expert.

**Table 3 healthcare-09-00665-t003:** Qualitative analysis of the assessment of usual practice in the case of there being clinical trials in progress in your hospital. Galicia, 2020.

	Questions	Qualitative Textual Quote
**General characteristics**	6. Do you receive financial or non-financial (academic or other) compensation for including patients in clinical trials? Specify the type of compensation.	*“There is only really academic compensation if you are one of the coordinators or the main recruiters. As far as financial compensation is concerned, I do not receive it directly; part of it goes to the hospital foundation (I do not know how whether it really indirectly benefits me) and part of it goes to the Oncology Unit to feed back into the trials by paying the data managers, although there are clearly not enough of them, and a small part is up to the decision of the head of the research unit.”* *(T7).*
	7. Have you identified whether the risks of clinical trials in which you have taken part are often greater than the benefits that the patient may receive? Explain what you have observed.	No complementary qualitative analyses have been performed.
**Logistics for handling clinical trials**	8. Are some of the medications identified or used in clinical trials currently used in your clinical practice?	*“I usually only take part in studies to which I have access and that provide a possible benefit over existing treatments or a new opportunity to compare with a placebo in advanced situations in patients that could potentially benefit from this. If I do not have a trial with these premises, I look for one outside. Being able to offer it in the patient environment and take part in it is an important professional incentive for me.”* *(T8).* *“Although it is not the case for all of them, the way to generalise the advantages of a new treatment or rule it out is through clinical trials; it is also an advantage to be able to offer your patients the option of benefiting from a drug in research when you do not have another truly effective treatment for them.”* *(T9).*
	9. What is the interest in or need for conducting a clinical trial in your hospital?	*“[Clinical trials] have provided a healthcare paradigm shift with quicker access to drugs by the participants; they have also implemented knowledge in specific clinical situations (biomarkers, fragility, advanced age, etc.); so I think they are important for patients.”* *(T10).*
	10. Do you consider that the monitors of clinical trials are sufficiently trained to control the characteristics of the clinical trials or have you observed that there was not adequate control? Explain your answer.	*“I am not sure, since the monitoring process is usually handled by the data manager team.” (T11).* *“In some cases, but not all, I have noticed a lack of adequate control by the monitors.”* *(T12).* *“Increasingly more monitors are poorly paid and have little training in clinical trials and the discipline concerned. They are normally university graduates who cannot find a job and decide to temporarily work in monitoring. Those that build up experience are promoted and inexpert people again do the monitoring … Of course, one cannot generalise about all of them.” (T13).* *“In recent years, the training of monitors has worsened. Changes, often during the trial itself, are detrimental to its monitoring.” (T14).*
	11. Does your centre have a clinical trial management unit or a data manager?	*“The oncology unit in my centre has a data manager and nursing [staff] paid for by the unit itself. The hospital does not provide us with any management unit resources ⋯ basically we are charged for everything we request from the hospital with no kind of advantage for the oncology unit.”* *(T15).*
	12. Is there any kind of collaboration with external groups so that, once the clinical trial has been performed, the information can be analysed?	*“We essentially work with cooperative research groups or with the industry, which handles data analysis” (T16).* *“Most the time it is external companies that perform the analysis and monitoring. Cooperative groups (I belong to several of them), hire the external CROs (contract research organisations), which handle data collection and analysis.” (T17).*
	13. Have you observed whether, when an important clinical trial is performed in your hospital, there are improved logistics for monitoring equipment or treatment equipment that will be subsequently used for patients? What equipment?	No complementary qualitative analyses have been performed.
	14. If there is, or already was, new equipment donated by pharmaceutical companies, new units implemented by pharmaceutical companies or any additional benefit for the hospital, do you consider that receiving these benefits conditions the recruitment or inclusion of patients in the study?	*“We try to carry out active recruitment under any conditions, with or without an additional benefit for the hospital.”* *(T18).*
	15. Does the patient’s socio-economic situation influence his/her participation in their study?	*“It should not. The inclusion of the patient in a clinical trial must depend on whether they meet the inclusion criteria and no exclusion criteria, that they understand the objective of the trial and undertake to comply with the trial’s demands. If the patient’s socio-economic situation is negatively affected by inclusion in a clinical trial, he/she may not stick with it. And if this is repeated, it should be considered whether the clinical trial should cover the patient’s costs.”* *(T19).*
	16. Do the number of tests that must be performed and patients’ comorbidities influence recruitment for a clinical trial?	*“Yes, this does have an influence due to the personal impact and the need to travel to the hospital.”* *(T20).* *“Yes, it does influence patients who are potentially vulnerable, highly symptomatic, who have to wait, etc.”* *(T21).*
**Participants selection**	17. Do you think that the financial or academic benefit received for patients included in the study affected any of your criteria for including a patient?	*“I do not include more patients with financial/academic compensation in mind. If I think the study is of interest to clinical practice and I have committed myself to taking part, I try to include the patients I see that meet the criteria. I should point out that we even take part in some studies that do not cover the costs but are academically useful (we offset this with money left over from other studies).”* *(T22).*
	18. Do you consider that sometimes you have been forced to include patients in a clinical trial because there are no other treatment alternatives in your hospital?	*“There are treatments used in a clinical trial that are not available in my centre, so if there is no treatment alternative, they are referred to a centre where it is possible to take part in a clinical trial.”* *(T23).* *“A clinical trial may be an alternative for a patient that can no longer receive healthcare medications, but they are not forced to do it, it is seen as a good option for that patient.” (T24).*
	19. When you recommend a patient to take part in a study, are you convinced that it is the most beneficial therapeutic option for the patient?	*“I always look out for the patient’s benefit.” (T25).* *“Normally, yes, since if I think I have another valid and better option, I request it or use it as treatment. When there is a trial in progress we have to at least offer it to the patient as an option, but if they ask me for my opinion about other different options, I am frank about it.”* *(T26).*
	20. Regarding the criteria for including the patient in the study, select the correct option: (a) It is difficult to find the ideal patient. (b) You find it hard to recruit patients who meet all of the criteria; is not easy to get them to accept. (c) Sometimes, this results in the same patient taking part in successive studies. (d) They are all correct. (Multi-reponse question)	No complementary qualitative analyses have been performed.
**Doctors’ motivation**	21. Do you think it is important to participate as an author of the clinical trials on which you have collaborated?	*“I think it is beneficial for researchers themselves in terms of their CVs although I do not think it is the main objective for taking part in research.”* *(T27).* *“Due to recognition of the work performed.”* *(T28).* *“The effort and the work must be seen to be rewarded.”* *(T29).*
	22. Is it a requirement for you to be allowed to be an author in order to participate in a clinical trial?	*“It is not a priority objective for me. The main objective for taking part in trials should not be that it improves your CV. I consider that to be an indefensible perverse objective.”* *(T30).* *“The important thing is to offer something more to our patients.”* *(T31).*
	23. Before taking part in the first clinical trial, had you issued publications?	No complementary qualitative analyses have been performed.
	24. As a medical professional, do you think that the financial incentive influences the decision to take part in a clinical trial?	*“A clinical trial involves extra work and financial remuneration is a way of paying for that work.” (T32).* *“At my centre, we have to pay lots of bills, including research personnel salaries; if the trial does not cover the expenses, it is very difficult to take part.”* *(T33).* *“Not really, because I do not receive the benefit and because the clinical benefit of my patients always takes priority, although sometimes it is true that considerable payment is made (normally in proportion to the amount of work it involves), which may benefit the unit mainly in terms of an increase in research staff.”* *(T34).*
	25. Do you think the clinical trials in progress will be useful for implementation in the future?	*“Obviously not all of them, but many of them, if they are positive, would offer new therapeutic options to patients and others clear up doubts about how to handle neoplasms without that meaning a new drug per se.”* *(T35).* *“The treatments employed in clinical trials could be highly useful if they prove to have benefits or provide data about treatment sequences or subpopulations with a greater benefit.”* *(T36).*

T: testimony. The testimonials (T) represent the opinions of the experts; more than one opinion may have been issued by an expert.

**Table 4 healthcare-09-00665-t004:** Qualitative analysis of knowledge of clinical trials. Galicia, 2020.

Questions	Qualitative Textual Quote
27. Do you know how biological markers are used in clinical trials?	No complementary qualitative analyses have been performed.
28. Do you know any kinds of biases that may affect a clinical trial?	No complementary qualitative analyses have been performed.
29. Which part of the clinical trial do you analyse before deciding to take part in a study?	No complementary qualitative analyses have been performed.
30. Do you consider yourself capable of analysing a clinical trial and deciding if it is well-produced or has biases?	No complementary qualitative analyses have been performed.
31. Before you start prescribing new drugs, do you analyse the clinical protocols to find out whether the population is similar to the one being treated or if the study has been well-produced methodologically?	No complementary qualitative analyses have been performed.
32. Have you identified any groups that have exerted pressure for the use of a particular treatment?	*“I feel pressure from the industry and, on occasions, from prestigious oncologists.”* *(T37).* *“The pharmaceutical industry always applies pressure, with greater or lesser subtlety, in order to increase the prescribing of new drugs. Our duty as oncologists is to be critical and maintain an appropriate distance.”* *(T38).*

The testimonials (T) represent the opinions of the experts; more than one opinion may have been issued by an expert.

## Data Availability

The datasets used and/or analysed during the current study are available from the corresponding author on reasonable request.

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
