# Peer review of "Do Clinical Trials Meet Current Care Needs? Views of Digestive Oncology Specialists in Galicia (Spain) Using the Delphi Method"

_healthcare, 2021, doi:10.3390/healthcare9060665_

Round 1

Reviewer 1 Report

Dear authors,

The introduction is something concise, but overall is fine. With respect to methodology, it would be appropriate to consider some suggestions.

First, on line 90, th I suggest the authors refer to the prospective study using the Delphi method or technique as an adapted or modified study, for example "a modified Delphi method". Likewise, it would eliminate the use of a simple-round, since in the initial design more than one round must be contemplated. 

Second, regarding the identification of the participants, at the beginning, line 108 would be convenient to include that it was carried out by  intentional non-probabilistic sampling. 

Also, in line 178, the word Consensus does not make sense in the sentence. 

I think the study was done in two rounds. The first round began with the development of the questionnaire. The experts were asked to identify the questions. In a second round, they evaluated the questions. They reached consensus (64%). Therefore, the rounded number would have to be renumbered, on line 180, 206, 208 and 223.

In line 214, eliminate "the sample was saturated", because the information analyzed was quantitative.

I suggest including a Consensus section. Include here the quantitative and qualitative criteria.

Were the final results transmitted to the participants? Something should be included on this matter. 

The results and the discussion are consistent. 

I suggest some references:

McPherson, S., Reese, C., & Wendler, M. C. (2018). Methodology update: Delphi studies. Nursing research, 67(5), 404-410

Barrett, D., & Heale, R. (2020). What are Delphi studies?. Evidence-Based Nursing, 23(3), 68-69.

Author Response

Reviewer 1

Dear authors,

The introduction is something concise, but overall is fine. With respect to methodology, it would be appropriate to consider some suggestions.

Rev 1 Observation 1:

First, on line 90, I suggest the authors refer to the prospective study using the Delphi method or technique as an adapted or modified study, for example "a modified Delphi method". Likewise, it would eliminate the use of a simple-round, since in the initial design more than one round must be contemplated. 

Response:

We have changed the text, accepting the reviewer's recommendation:

The text says:

Combined (qualitative and quantitative), prospective study using the single-round Delphi method.

A several-round Delphi study was proposed. However, due to information saturation in the first round, the study was completed in a single round

The text should say:

Combined (qualitative and quantitative), prospective study using a modified Delphi method. A several-round Delphi study was proposed. However, due to information saturation in the second round, the study was completed.

Rev 1 Observation 2:

Second, regarding the identification of the participants, at the beginning, line 108 would be convenient to include that it was carried out by intentional non-probabilistic sampling. 

Response:

We have included the following information in Stage 1 under the sampling heading:

Sampling: Intentional non-probabilistic sampling was used.

Rev 1 Observation 3:

Also, in line 178, the word Consensus does not make sense in the sentence. 

Response: We have removed the word consensus from this part and created a section with that title.

Rev 1 Observation 4:

I think the study was done in two rounds. The first round began with the development of the questionnaire. The experts were asked to identify the questions. In a second round, they evaluated the questions. They reached consensus (64%). Therefore, the rounded number would have to be renumbered, on line 180, 206, 208 and 223.

Response: We have reorganised the part concerning Stage 2 with the following text:

Stage 2: Type of analysis and consensus:

The questionnaire, made up of closed and open questions, made it possible to conduct a descriptive analysis (quantitative method) and a discourse analysis (qualitative method).

Selection of questions for the test:

First Round: this was carried out with the RiGhT-sens working group. The RIGhT-sens group was shown each of the probable questions together with explanatory text. They were then asked to vote to maintain, remove or modify the question, or state that they had no opinion. We used categorical response options to ensure that the people taking the survey fully understood the consequences of their votes, to clarify the interpretation and to ensure that the results could be processed in such a way as to establish a final list of questions at the end of the study. Consensus

Second Round: In the second round, two types of qualitative and quantitative analysis were carried out, as described below:

Quantitative analysis: in order to analyse the first round, SPSS statistical software was used to perform a descriptive analysis of the closed-ended answers (questions in which an alternative was selected).

Qualitative analysis: In order to supplement the descriptive information, a discourse analysis was conducted from which literal expressions were extracted, which are included in the results report. analysis of the contents of the qualitative questions was carried out..

2.4. Second round test

Once the questions had been selected, a link to the tests was sent. All of the questions had options to reduce the response margin and a free text option to add supplementary information if it was necessary to add it.

Consensus: The frequency distribution of each variable was then calculated and the highest percentage for each answer was identified.

In a second stage, the MODE of the highest percentage of all answers was calculated in order to establish a consensus value, which was finally set at 64,29% for the majority of variables analysed. We chose the mode in order to represent the highest number of responses and prevent our observations being affected by extreme values. When we verified the value established, we also confirmed that the consensus value is within the ranges proposed in other research, which mentions that a consensus level can vary from 51% to 100% and that a level of 100% is unlikely to be reached[1].[2]

There was considered to be consensus when there was more than 64% agreement with the identified responses. For questions with a lower level of consensus, the analysis was supplemented complemented with open text information qualitative analysis (content analysis) in accordance with any of the available actions. The RIGhT-sens teams analysed the responses to the round 1 2 questionnaire. "No opinion" responses were excluded from the agreement percentage calculations. The initial plan was that whenever there was no consensus on maintaining or removing a question, it would not be included in the questionnaire in round 2 3. However, such discrimination was not necessary since there was not sufficient information for a second third round. In cases in which the panel did not reach at least 64% agreement to maintain or remove a question, we examined the comments and extrapolated the most important idea.

After the initial analysis, it was identified that a high percentage of questions reached a consensus level higher than 64%, so it was decided that the sample was saturated and that it was not necessary to conduct a second round. The responses with a lower consensus percentage were supplemented with qualitative information analysis provided in open questions. 

The contents were analysed in parallel by two researchers (an epidemiologist and a health anthropologist) in order to avoid intersubjectivity and triangulate the results. Once it had been collected, the information was analysed until the discourse was saturated. Consensus concerning the results and their internal consistency made it unnecessary to perform a second third round of consultation.

Comunicación de la información a los participantes:

All of the information was shared with the working group and the minimum consensus was established with them.

All of the information was shared with the participants once the analysis had been completed.

Rev 1 Observation 5:

In line 214, eliminate "the sample was saturated", because the information analyzed was quantitative.

I suggest including a Consensus section. Include here the quantitative and qualitative criteria.

Response: we have removed the information: "the sample was saturated", based on the reviewer's suggestion, and added a section called consensus.

Rev 1 Observation 6:

Were the final results transmitted to the participants? Something should be included on this matter. 

Response: The information was shared with the participants and we have also included a section stating that the information has been conveyed to the participants:

The text says: All of the information was shared with the working group and the minimum consensus was established with them.

It should say: All of the information was shared with the participants once the analysis had been completed.

Rev 1 Observation 7:

The results and the discussion are consistent. 

Response: Thank you for your comment.

Observation 8:

I suggest some references:

McPherson, S., Reese, C., & Wendler, M. C. (2018). Methodology update: Delphi studies. Nursing research, 67(5), 404-410

Barrett, D., & Heale, R. (2020). What are Delphi studies?. Evidence-Based Nursing, 23(3), 68-69.

Response: We have included the references in accordance with the reviewer's suggestions.

[2] Barrett D, Heale R. What are Delphi studies? Evid Based Nurs. 2020 Jul;23(3):68-69. doi: 10.1136/ebnurs-2020-103303. Epub 2020 May 19. PMID: 32430290

Reviewer 2 Report

Firstly I would like to issue praise for their interest in tackling the issues of the point of view of the clinical tries, and their practical use, as outlined in the article. Personally, I think that the subject is of great importance and that more work should be done in this field.

Regarding the article, I would like to make the following comments to improve the message they intend to convey:

In general the article seems very good.

The subject is novel and important.

1.- I suggest that the introduction the authors explain a little more about the RWD studies,  I believe  that this issue it is a relevant point that should be highlighted in the discussion and in the conclusions.

2.- The other hand the authors should include in the  discussion: the importance of have in the hospitals a statistics team or expert committee which will give support to the professionals during clinical tries and they can help to review evidence-based protocols.

3.- As conclusions, the authors should include this affirmation: This study evidence that some times make a decisions in hospitals are made individually by each oncologiest specialist and not always using established protocols.

Author Response

Reviewer 2

Firstly I would like to issue praise for their interest in tackling the issues of the point of view of the clinical tries, and their practical use, as outlined in the article. Personally, I think that the subject is of great importance and that more work should be done in this field.

Regarding the article, I would like to make the following comments to improve the message they intend to convey:

In general the article seems very good.

The subject is novel and important.

Rev. 2 Observation 1:

1.- I suggest that the introduction the authors explain a little more about the RWD studies,  I believe  that this issue it is a relevant point that should be highlighted in the discussion and in the conclusions.

Response: We have included information in the introduction that explains the definition of RWD studies.

the term real- world data (RWD) refers to population- level data obtained from cancer registries and not patient information extracted from a study[i]. The Food and Drug Administration (FDA) has defined RWD as data relating to patient health status and/or care records that are real and that are collected during the patient's care under real rather than ideal conditions. This information comes from the electronic clinical history and from administrative records that the patient was added to during clinical care[ii].

Rev. 2 Observation 2:

2.- The other hand the authors should include in the discussion: the importance of have in the hospitals a statistics team or expert committee which will give support to the professionals during clinical tries and they can help to review evidence-based protocols.

Response: We have included the following information in the final part of the discussion:
It is therefore important for hospitals to have a research department, which should have a statistical team or committee of experts who are responsible for helping and supporting people interested in conducting research with cancer patients.

3.- As conclusions, the authors should include this affirmation: This study evidence that sometimes make a decisions in hospitals are made individually by each oncologiest specialist and not always using established protocols.

Response: We have included a conclusion that states:

This study shows that, often, the decisions in some situations are decisions that should be individualised for each patient because no protocols are identified that are adjusted to such specific situations at the time of treating the oncology patient.

[i] Goldberg, R. M., Wei, L. & Fernandez, S. The evolution of clinical trials in oncology: defining who benefits from new drugs using innovative study designs. Oncologist 22, 1015–1019 (2017).

[ii] Sherman, R. E. et al. Real- world evidence — what is it and what can it tell us? N. Engl. J. Med. 375, 2293–2297 (2016).

Reviewer 3 Report

It is a pleasure to be able to participate in the revision and improvement of this article. The theme is very interesting and relevant.

The main questions that I raise to the authors before any more detailed analysis, concern the methodological approach used.

In my opinion, in order to study the perception of medical specialists in oncology, a qualitative approach should be used, namely through content analysis.

The Delphi technique is a methodology for reaching consensus among a group of experts. In the present study, the panel of experts was asked to answer 32 questions.

The step 2: “Type of analysis and consensus:” is not very well perceptible namely:

  • Line 185: “In a second stage, the average of the highest percentage of all answers was calculated in order to establish a consensus value, which was finally set at 64% for the majority of variables analyzed.” - How was the consensus obtained from open questions? With what criteria was this value (64%) established?
  • In Line 188: “In order to supplement the descriptive information, a discourse analysis was conducted from which literal expressions were 190 extracted, which are included in the results report.” - In my view, this is content analysis.
  • In Line 191: “All of the information was shared with the working group and the minimum consensus was established with them.” - what has been accomplished is not noticeable.

I do not understand how to calculate agreement based on open questions. The data present in tables 2 and 3 have illustrated: “qualitative textual quote”. That is typical of qualitative approaches such as content analysis.

The results they find are interesting, but I do not agree with the methodological approach that the authors take at the instruments used and data obtained.

The limitations presented are typical of other types of studies other than this one. Statistical inference is not a priority in qualitative approaches.

Author Response

Reviewer 3

Rev 3 Comment 1:

It is a pleasure to be able to participate in the revision and improvement of this article. The theme is very interesting and relevant.

The main questions that I raise to the authors before any more detailed analysis, concern the methodological approach used.

In my opinion, in order to study the perception of medical specialists in oncology, a qualitative approach should be used, namely through content analysis.

The Delphi technique is a methodology for reaching consensus among a group of experts. In the present study, the panel of experts was asked to answer 32 questions.

Response: Thank you for your comments.

Rev 3 Observation 1:

The step 2: “Type of analysis and consensus:” is not very well perceptible namely:

  • Line 185: “In a second stage, the average of the highest percentage of all answers was calculated in order to establish a consensus value, which was finally set at 64% for the majority of variables analyzed.” - How was the consensus obtained from open questions? With what criteria was this value (64%) established?

Response: We have reformulated the information on the methodology in response to the observations made by reviewers 1 and 2 and the observation you kindly sent us.

In response to the suggestions made by reviewer 1, who evaluated the methodology, we have considered the consultations and drafting of the first questions (work by RIGhT-sens group experts) as the first round and the evaluation with the group of experts consulted as the second round.

We have thus created a section in which we explain how consensus was obtained and how the rounds were carried out. In response to an observation made by reviewer 1 we have created a section called consensus in which we have provided details of how the consensus was reached. We apologise for a Spanish-English translation error, which resulted in the average being used to determine the consensus value when the MODE was actually used. These terms are written similarly in Spanish. We have corrected this error and attach the modified text.:

Consensus: The frequency distribution of each variable was then calculated and the highest percentage for each answer was identified.

In a second stage, the MODE of the highest percentage of all answers was calculated in order to establish a consensus value, which was finally set at 64,29% for the majority of variables analysed. We chose the mode in order to represent the highest number of responses and prevent our observations being affected by extreme values. When we verified the value established, we also confirmed that the consensus value is within the ranges proposed in other research, which mentions that a consensus level can vary from 51% to 100% and that a level of 100% is unlikely to be reached.3,4

There was considered to be consensus when there was more than 64% agreement with the identified responses. For questions with a lower level of consensus, the analysis was supplemented complemented with open text information qualitative analysis (content analysis) in accordance with any of the available actions. The RIGhT-sens teams analysed the responses to the round 1 2 questionnaire. "No opinion" responses were excluded from the agreement percentage calculations. The initial plan was that whenever there was no consensus on maintaining or removing a question, it would not be included in the questionnaire in round 2 3. However, such discrimination was not necessary since there was not sufficient information for a second third round. In cases in which the panel did not reach at least 64% agreement to maintain or remove a question, we examined the comments and extrapolated the most important idea.

After the initial analysis, it was identified that a high percentage of questions reached a consensus level higher than 64%, so it was decided that the sample was saturated and that it was not necessary to conduct a second round. The responses with a lower consensus percentage were supplemented with qualitative information analysis provided in open questions.

Rev 3 Observation 2:

  • In Line 188: “In order to supplement the descriptive information, a discourse analysis was conducted from which literal expressions were 190 extracted, which are included in the results report.” - In my view, this is content analysis.

Response: we have corrected the text in accordance with the reviewer's suggestions.

The text says:

In order to supplement the descriptive information, a discourse analysis was conducted from which literal expressions were extracted, which are included in the results report.

It should say:

Qualitative analysis: In order to supplement the descriptive information, a discourse analysis was conducted from which literal expressions were extracted, which are included in the results report. Analysis of the contents of the qualitative questions was carried out.

2.4. Second round test

Once the questions had been selected, a link to the tests was sent. All of the questions had options to reduce the response margin and a free text option to add supplementary information if it was necessary to add it.

Rev 3 Observation 3:

  • In Line 191: “All of the information was shared with the working group and the minimum consensus was established with them.” - what has been accomplished is not noticeable.

Response: We have modified this paragraph and we have also included a section stating that the information has been shared to the participants:

The text says: All of the information was shared with the working group and the minimum consensus was established with them.

It should say: All of the information was shared with the participants once the analysis had been completed.

Rev 3 Observation 4:

I do not understand how to calculate agreement based on open questions. The data present in tables 2 and 3 have illustrated: “qualitative textual quote”. That is typical of qualitative approaches such as content analysis.

The results they find are interesting, but I do not agree with the methodological approach that the authors take at the instruments used and data obtained.

Response: We think that the way in which we have explained the consensus can be improved considerably, so we have made changes to the methodology. These changes include incorporating the first round (suggested by reviewer 1), the second round, and incorporating a quantitative analysis section, a qualitative analysis section and a consensus section (the last three at your suggestion). These changes can be seen in detail in the new corrected text. We hope these changes will contribute to improving the information we are seeking to convey.

Stage 2: Type of analysis and consensus:

The questionnaire, made up of closed and open questions, made it possible to conduct a descriptive analysis (quantitative method) and a discourse analysis (qualitative method).

Selection of questions for the test:

First Round: this was carried out with the RiGhT-sens working group. The RIGhT-sens group was shown each of the probable questions together with explanatory text. They were then asked to vote to maintain, remove or modify the question, or state that they had no opinion. We used categorical response options to ensure that the people taking the survey fully understood the consequences of their votes, to clarify the interpretation and to ensure that the results could be processed in such a way as to establish a final list of questions at the end of the study. Consensus

Second Round: in the second round, two types of qualitative and quantitative analysis were carried out, as described below:

Quantitative analysis: in order to analyse the first round, SPSS statistical software was used to perform a descriptive analysis of the closed-ended answers (questions in which an alternative was selected).

Qualitative analysis: In order to supplement the descriptive information, a discourse analysis was conducted from which literal expressions were extracted, which are included in the results report. analysis of the contents of the qualitative questions was carried out.

2.4. Round Second test***********

Once the questions had been selected, a link to the tests was sent. All of the questions had options to reduce the response margin and a free text option to add supplementary information if it was necessary to add it.

Rev 3 Observation 4:

The limitations presented are typical of other types of studies other than this one. Statistical inference is not a priority in qualitative approaches.

Response: We have modified the text on limitations in accordance with your suggestion:

The text says:

Due to the small size of the population studied, a larger study will be required, with national coverage, in order to be able to generalise these results.

It should say:

Despite the small size of the population studied, and despite the fact that it would seem intuitive that a larger study might be required to identify a sample from areas of Spain other than Galicia, we are aware that, from a qualitative point of view, causal inference is not an important limitation. We therefore consider that these findings are valid and could be very useful for other oncologists both in Spain and around the world. Moreover, qualitative research allows us to identify findings that it would be difficult to demonstrate from a quantitative point of view.

Round 2

Reviewer 3 Report

I thank you in advance for your responses to my comments.

But I still have a big question about the methodology:

  1. The working group (RIGhT-sens group) elaborated the questions to be evaluated? - This aspect should be made clearer in the text.
  2. Oncology experts were invited to answer this questionnaire. This is a different group than the first, right?
  3. In view of this, I cannot consider that was made two rounds, as you have 2 different groups.
  4. Was the Delphi methodology used to obtain consensus in the realization of the questions in the initial questionnaire? Or was it used in the responses to the questionnaire? There is a lot of confusion in the text regarding this aspect. This is not clear in de text.

Another aspect: 14 experts were invited to answer the questionnaire. Why are qualitative citations numbered up to T28?

I apologize for this.

Author Response

Dear Editors;

Thank you for your comments. As the authors, we trust that we have been able to correct each of the observations that you have so kindly made.

We would be grateful if you could review this latest version, which we trust better meets the criteria you have stated.

The changes made are detailed below:

  1. The working group (RIGhT-sens group) elaborated the questions to be evaluated? - This aspect should be made clearer in the text.

Response:

The working group, which was made up of a different group of oncologists from the group in the second round, was entrusted with preparing a list of questions. However, since there was a very high number of questions, an analysis of the questions was performed. We have modified the manuscript after responding to your suggestions under the questionnaire preparation title.

  1. Oncology experts were invited to answer this questionnaire. This is a different group than the first, right?

Response:

Yes, the oncologists who were part of the expert group are a different group from the working group.

  1. In view of this, I cannot consider that was made two rounds, as you have 2 different groups.

Response:

In response to reviewer 2's suggestion in the previous review, we changed the text to include a second round. However, taking into account their point of view, we consider their suggestion appropriate and we believe that it is more consistent with the work carried out by our team, so we have modified the entire text and stated that it was performed in a single round, as we initially expressed in the original text sent the first time. Thank you very much for this suggestion. We think it will be a great help in avoiding confusion.

  1. Was the Delphi methodology used to obtain consensus in the realization of the questions in the initial questionnaire? Or was it used in the responses to the questionnaire? There is a lot of confusion in the text regarding this aspect. This is not clear in the text.

Response:

As we responded to question 1, in the first stage the working group produced the questionnaire and the questions were tested with the members of the working group. According to reviewer 2's suggestion, that would constitute the first round. However, we have modified the wording and left the original text.

Another aspect: 14 experts were invited to answer the questionnaire. Why are qualitative citations numbered up to T28?

The list is the number of testimonials issued by the experts. For some experts, we cited more than one testimony.

We have included a clarifying note at the end of each table so as not to create confusion.

The note says:

The testimonials (t) represent the opinions of the experts; more than one opinion may have been issued by an expert.
